# Characterization of the Habitat- and Season-Independent Increase in Fungal Biomass Induced by the Invasive Giant Goldenrod and Its Impact on the Fungivorous Nematode Community

**DOI:** 10.3390/microorganisms9020437

**Published:** 2021-02-19

**Authors:** Paula Harkes, Lisa J. M. van Heumen, Sven J. J. van den Elsen, Paul J. W. Mooijman, Mariëtte T. W. Vervoort, Gerrit Gort, Martijn H. M. Holterman, Joris J. M. van Steenbrugge, Casper W. Quist, Johannes Helder

**Affiliations:** 1Laboratory of Nematology, Wageningen University, Droevendaalsesteeg 1, 6708 PB Wageningen, The Netherlands; lisaheumen@gmail.com (L.J.M.v.H.); sven.vandenelsen@wur.nl (S.J.J.v.d.E.); Paul.mooijman@wur.nl (P.J.W.M.); jet.vervoort@wur.nl (M.T.W.V.); martijn.holterman@solynta.com (M.H.M.H.); Joris.vansteenbrugge@wur.nl (J.J.M.v.S.); Casper.quist@wur.nl (C.W.Q.); Hans.helder@wur.nl (J.H.); 2Soil Physics and Land Management Group, Wageningen University, Droevendaalsesteeg 4, 6708 PB Wageningen, The Netherlands; 3Bioscience, Wageningen Plant Research, Droevendaalsesteeg 1, 6708 PB Wageningen, The Netherlands; 4Biometris, Wageningen University, Droevendaalsesteeg 1, 6708 PB Wageningen, The Netherlands; gerrit.gort@wur.nl; 5Solynta, Dreijenlaan 2, 6703 HA Wageningen, The Netherlands; 6Biosystematics Group, Wageningen University, Droevendaalsesteeg 1, 6708 PB Wageningen, The Netherlands

**Keywords:** *Solidago gigantea*, invasion ecology, fungal community, 18S sequencing

## Abstract

Outside its native range, the invasive plant species giant goldenrod (*Solidago gigantea*) has been shown to increase belowground fungal biomass. This non-obvious effect is poorly characterized; we don’t know whether it is plant developmental stage-dependent, which fractions of the fungal community are affected, and whether it is reflected in the next trophic level. To address these questions, fungal assemblages in soil samples collected from invaded and uninvaded plots in two soil types were compared. Although using ergosterol as a marker for fungal biomass demonstrated a significant increase in fungal biomass, specific quantitative PCR (qPCR) assays did not point at a quantitative shift. MiSeq-based characterization of the belowground effects of giant goldenrod revealed a local increase of mainly Cladosporiaceae and Glomeraceae. This asymmetric boost in the fungal community was reflected in a specific shift in the fungivorous nematode community. Our findings provide insight into the potential impact of invasive plants on local fungal communities.

## 1. Introduction

Invasive plants pose a threat to the biodiversity of natural ecosystems [1]. Aboveground observations have often revealed that invasive plants reduce species richness of the native plant community [2]. Belowground, invasive plants can change physical conditions and the composition of soil biota, thereby even negatively affecting the soil biological conditions for the native plant community, making the restoration of the original vegetation more difficult [3]. Although invasion can change ecological links, it is still unclear how invasive plant species modify soil community composition, diversity, structure, and function [4]. Understanding the mechanisms contributing to the success of plant invasion is critical for ecosystem conservation [5]. Therefore, we need to identify the community- and ecosystem-level effects of the invasive species and discover which traits enable its invasive success [6].

*Solidago gigantea* is a rhizomatous perennial herb, native to North America [7]. It was introduced in Europe as an ornamental plant and rapidly became invasive in Europe and parts of Asia, quickly becoming one of Europe’s most detrimental plant invaders [8]. In North America, *S. gigantea* prefers wetlands, whilst in Europe it can survive under a broad range of light intensities, soil moistures, temperatures, nutrient conditions, and pH [9]. In its native habitat, *S. gigantea* is colonized by mycorrhizal fungi [10]. Zubek et al. showed that giant goldenrod also interacted with arbuscular mycorrhizal fungi (AMF) outside its native range, and the AMF colonization rates were higher in invaded plots as compared to neighbouring, uninvaded plots [11]. Additionally, shared mycorrhizal networks between plants showed an increased growth and nutrient acquisition by invasive *Solidago canadensis* at the expense of the native plant *Kummerowia striata* [12]. AMF have also been shown to stimulate the decomposition of plant material [13,14,15]. Although the exact mechanisms of this effect are unknown, it is hypothesised that AMF export nitrogen from litter and slowly release low concentrations of labile C, thereby altering the relative abundance of saprotrophic microbial communities in soil [15,16]. These observations show that plant mycorrhizal status might be linked to the success of plant invasion [17,18]. Since *S. gigantea* is a rhizomatous perennial herb that forms nearly pure monoculture stands in various habitats, giant goldenrod is an appropriate species to use to study the effect of invasive plants on soil biota. 

In previous studies on the belowground effects of the invasive giant goldenrod, a local increase in the overall fungal biomass was detected, both in a mesocosm experiment [10] and under semi-natural conditions [19,20]. The total fungal biomass was assessed using ergosterol, a biochemical marker for higher fungi, or PLFA 18:2ω6. Ergosterol is a valid marker for major fungal groups such as Ascomycota and Basidiomycota, but it should be noted that some fungal groups such as the Glomeromycota—which form arbuscular mycorrhizas—and the Chytridiomycota lack this sterol in their cell membranes [21]. On the other hand, PLFA 18:2ω6 has been found in various plant species. Each method to estimate fungal biomass comes with its own limitations. 

Invasive plant-induced changes in the fungal community might also be mirrored in the next trophic level such as fungivorous metazoan communities. Fungivorous nematodes are informative in this context as they are present in nearly all soil habitats. Their ability to feed on fungi evolved multiple times independently [22], which has resulted in lineages with distinct food preferences [23,24]. 

Here, we investigated the impact of invasive *S. gigantea* on local fungal communities in more detail. First, we verified whether the *S. gigantea*-induced increase in fungal biomass was transient or long-lasting. To confirm this, soil samples were collected at the end of the growing season (November). Second, ribosomal DNA-based markers were used along with ergosterol to characterize changes in the fungal community. The use of two independent markers for fungal biomass provided a more solid basis for our findings, and fungal division-specific markers allowed us to characterize the impact qualitatively. To further characterize the fungal community, ribosomal DNA amplicons were sequenced. In addition, we checked whether representatives of the next trophic level, fungivorous nematodes, were affected. Two out of the three nematode lineages present in these sites were stimulated in the presence of giant goldenrod. Possible explanations for these interesting but diverse results are discussed.

## 2. Materials and Methods

### 2.1. Sampling Sites 

The belowground effects of *S. gigantea* invasions were examined at eight sites in the Netherlands, located in either of the distinct semi-natural habitats, namely riparian zones (rive clay soil) and semi-natural grasslands (sandy soil) (see Table 1). For all invaded plots, the coverage by *S. gigantea* was scored as a 9 on a modified Braun-Blanquet scale [25,26], implying a 75–100% coverage. Uninvaded plots were dominated by native plant species, and if *S. gigantea* was present in the uninvaded plots, its coverage was neglectable. 

### 2.2. Soil Sampling 

In total, 104 composite soil samples were collected from 52 plot-pairs in November 2014. Four plot-pairs were sampled at sites due to the limited number of *S. gigantea* patches (see also Table 1). Each plot-pair consisted of two directly neighbouring 2 × 2 m plots to minimize possible differences in soil type and structure. To average microscale variation, 12 soil cores (depth: 25 cm, ø 1.5 cm) were randomly collected within each plot and mixed. Sampling material was thoroughly cleaned between plot-pairs in order to limit cross contamination. On the day of sample collection, composite soil samples were split into two subsamples (200 g and 5 g). The 200 g subsample was stored at 4 °C for subsequent nematode extraction and the determination of abiotic soil characteristics. The other subsamples (5g) for ergosterol and DNA extraction were stored at –20 °C to prevent DNA degradation. 

### 2.3. Nematode Extraction and Community Analysis

Nematodes were extracted from a 100 g subsample within one week after sample collection using an Oostenbrink elutriator [27]. DNA extractions of the total nematode suspensions were performed as described by Vervoort et al. [28]. At the start of this extraction procedure, 25 µL of calf thymus DNA (20.6 ng/µL) was added to each sample to be able to quantify DNA loss after extraction and purification. After purification, each sample was diluted 10 times and stored at −20 °C until further use. Diluted DNA extracts served as a template for the quantitative PCR (qPCR)-based determination of the total nematode density and the densities of the three major fungivorous nematode lineages present at these locations, namely Aphelenchidae, Aphelenchoididae, and *Diphtherophora*. This qPCR detection method is based on taxon-specific SSU rDNA sequence motifs as previously described [28].

### 2.4. Abiotic Soil Characteristics

Per composite sample, 60 g subsamples were taken for the analysis of abiotic and biotic soil characteristics. Moisture content, pH, organic matter (OM) content, total carbon (C) content, total nitrogen (N) content, and C:N ratio were determined. The total amount of C and N, determined with a composite sample from an invaded plot and an uninvaded plot per sampling site, was performed by BLGG AgroXpertus (Wageningen, The Netherlands).

Soil moisture content was measured for each sample by determining the sample’s weight loss after 20 h at 105 °C. Dried soil was sieved with a mesh of 2 mm and 10 g of dried soil was added to 25 mL demineralized water for soil pH measurements using a gel-electrolyte electrode (Sentix 21, WTW, Weilheim, Germany). Organic matter content was determined by measuring weight loss in 20 g of sieved soil after 5 h at 550 °C. 

### 2.5. Fungal and Bacterial Extraction and Community Analysis

Fungal and bacterial DNA was extracted from 0.25 g of subsamples using the PowerSoil DNA Isolation Kit (MO BIO Laboratories, Carlsbad, California, USA). Slight changes were made to the manufacturer’s protocol. PowerBead Tubes were placed in a Qiagen Tissue Lyser for 7 min instead of 10 min to compensate for the high shaking frequency (30 Hz); 20 µL of internal control DNA (20.6 ng/µL calf thymus DNA) was added to each sample in order to monitor DNA losses during extraction and purification. To further reduce the impact of soil-derived PCR-inhibiting components, purified lysates were diluted 100 times. Diluted purified lysates were stored at 4 °C until quantitative PCR (qPCR) analysis and sequencing. Undiluted purified lysates were stored at −20 °C. Microbial communities were analysed using qPCR assays targeting total fungi (ITS1F/5.8 s) and total bacteria (16S ribosomal RNA) as well as three fungal phyla: Ascomycota, Basidiomycota, and Chytridiomycota (based on taxon-characteristic ITS regions). qPCR primers, qPCR conditions, and slope and intercept values describing the relationship between Ct-values and concentration of target bacterial or fungal DNA (ng/µL) were determined according to Harkes et al. [29].

Due to the substantial variation estimation in rDNA copy numbers in fungi, using only the ITS marker might not be appropriate for all fungal clades. Therefore, two qPCR assays for single-copy protein coding genes were included in this study (beta-tubulin (tub2) and the translation elongation factor 1-alpha (tef1)) since these genes are supposed to be less variable and occur as a single copy in fungi [30]. 

### 2.6. Ergosterol Measurements

Ergosterol was extracted from 1 g of soil using the alkaline extraction method as described by [31]. In a mixture of alkaline methanol and cyclohexane, ergosterol accumulated in the cyclohexane phase. After phase separation, the cyclohexane was removed by evaporation, and ergosterol was re-dissolved in methanol. Subsequently, high-performance liquid chromatography (HPLC) with photodiode array detection was used to separate and quantify the ergosterol content of the samples as described by [32].

### 2.7. PCR Amplification and Sequencing of Fungal 16S rDNA

The variable V7-V8 of fungal 18S was utilized as a target for the analyses of Illumina 18S rDNA sequencing. To prepare the samples for sequencing, a twostep PCR procedure was followed as described in [33]. First, a locus-specific primer combination extended with an Illumina read area and the appropriate adapter were used to produce primary amplicons in triplicate for all samples. PCR 2 was then conducted on 40× diluted amplicons of PCR1 to attach the Illumina index and the Illumina sequencing adaptor. Randomly picked products of PCR 1 and 2 were checked on gel to ensure that amplification would be successful. Finally, all PCR products were pooled and sent for sequencing. Sequencing was done at Bioscience (Wageningen Research, Wageningen, The Netherlands) using the Illumina MiSeq Desktop Sequencer (2 × 250 nt paired-end sequencing) according to the standard protocols. The raw sequences were submitted to the NCBI Sequence Read Archive (SRA) database under study accession numbers PRJNA563313.

### 2.8. Combined Analysis of Abiotic Characteristics and Quantitative Biotic Data 

The impact of *S. gigantea* invasion on abiotic soil properties and the densities of nematodes, fungi, and bacteria was analysed using mixed linear models (PROC MIXED, SAS software system version 9.2) [34]. When residuals did not approximate normal distributions, transformed data were used. OM, total C, total N, nematode densities, and densities of fungi and bacteria were log-transformed. A constant of 0.1 was added prior to the log-transformation to bypass any zero values. This was done for Aphelenchidae, Aphelenchoididae, *Diphtherophora*, and Chytridiomycota. 

A split-plot design was used for all ten study sites, with sampling sites forming the main plots, associated with the factor habitat type (riparian vegetation or semi-natural grassland). Two subplots per plot pair were associated with the factor plant invasion. This design was represented in the mixed models with random effects for sites, plot-pairs, and individual plots forming the random part of the model. The main effects of habitat type, invasion, and interaction between both factors formed the fixed part of the model. Random effects for site, plot-pairs, and individual plots formed the random part of the model. In this way, the total error variance was split into variance components for sites, plot-pairs within sites, and for individual plots within plot-pairs. Regarding pH, the mixed model took into account that variances were different for riparian vegetation habitats and semi-natural grasslands (as was noticed from residual plots). Hypothesis tests (with F-test statistics) for the significance of the main effects of habitat type, invasion, and their interaction on the soil variables were performed. *p*-values < 0.05 were considered significant. Regardless of the outcome of hypothesis tests on interaction and main effects, comparisons between invaded and uninvaded plots were made per habitat type, using F-tests. The results were presented as (back transformed) 95% confidence intervals for the estimated mean responses of the soil variables (obtained from “least squares means” outputs) in invaded and uninvaded plots per habitat type. Moreover, invasion impacts on soil variables were presented as ratios between estimated means of invaded plots and uninvaded plots.

### 2.9. Bioinformatics Framework and Statistics

The composition of the fungal communities of the soil samples was analysed based on the sequencing data obtained from the Illumina MiSeq platform. Reads were sorted into the experimental samples according to their index combination, quality trimmed by BBDUK, and then merged via VSEARCH [35,36]. Unique sequences were then clustered at 97% similarity by using the usearch_global method implemented in VSEARCH and a representative consensus sequence per de novo OTU was determined [36]. The clustering algorithm also performed chimera filtering to discard likely chimeric OTUs with the UCHIME algorithm in de novo mode [37] implemented in VSEARCH. Sequences that passed quality filtering were then mapped to a set of representative consensus sequences to generate an OTU abundance Table. Representative OTU sequences were assigned to a taxonomic classification via BLAST against the Silva database (version 12.8). Sequences not belonging to fungi were discarded from the 18S fungal dataset. Low-abundance OTUs (those with an abundance of < 0.005% in the total dataset) were discarded [38] prior to analysis. Samples were transformed using Hellinger transformation for all downstream analyses.

To investigate the indicator taxa involved in the differences in fungal communities between invaded and non-invaded plots, a linear discriminant analysis effect size (LEfSe) was conducted in Microbiome Analyst [39] to explore the differential microbial populations at the family level [40]. A significance level of α ≤ 0.05 was used in this study.

## 3. Results

### 3.1. Changes in Abiotic Soil Characteristics upon S. gigantea Invasion

To gain insight into the abiotic environment of the *S. gigantea*-invaded sites, the soil moisture content, pH, OM content, total C content, total N content, and C:N ratio were analysed. Significant changes were observed in soil moisture content, pH, and OM content between *S. gigantea*-invaded and uninvaded plots in riparian and semi-natural grasslands sites (Table 2). In contrast, no differences were observed between invaded and uninvaded plots for the total C content, total N content, and the C:N ratio (Table 2 and Table 3, Figure 1).

Plots invaded by *S. gigantea* had a lower soil moisture content than uninvaded plots (F_1,50_ = 6.58, *p* = 0.0134; Table 2, Figure 1). This overall effect could mainly be attributed to the slightly lower moisture content of invaded plots in the riparian vegetation habitats (F_1,50_ = 5.79, *p* = 0.0199; Table 3, Figure 1). 

Riparian vegetation habitats and semi-natural grasslands differed significantly in pH (F_1,6_ = 92.22, *p* < 0.0001; Table 2). Riparian vegetation sites had a slightly alkaline soil with a pH of 7.5, while semi-natural grasslands had a moderately acidic soil with a pH of 5.6 (see Table 3 for 95% confidence intervals). Overall, invasive plants didn’t affect soil pH (Table 2, Figure 1). However, splitting the vegetation sites by habitat type showed that for both types, the pH was slightly lower in invaded plots. Only for riparian sites was this difference significant (F_1,50_ = 5.81, *p* = 0.0197; Table 3).

A significant interaction between invasion and habitat was found for OM content (F_1,50_ = 4.74, *p* = 0.0341), indicating that the effect of invasion was habitat type-dependent (Table 2). In semi-natural grasslands, *S. gigantea*-invaded plots had a higher OM content as compared to uninvaded plots (F_1,50_ = 8.12, *p* = 0.0063; Table 3, Figure 1), whereas no difference in OM content was detected between plot-pairs at riparian sites. It should be noted that determining OM content by weight loss as used here might have overestimated the OM% for the riparian samples with a high clay content [41]. 

### 3.2. Invasive S. Gigantea Increase Fungal Biomass, but Not the Total Fungal DNA 

Using ergosterol as a biochemical marker for biomass of higher fungi, a strong overall effect of *S. gigantea* was detected (F_1,48_ = 21.97, *p* < 0.0001; Table 2). In giant goldenrod-invaded plots, a significant increase in ergosterol levels was observed for both habitat types (Table 2 and Table 3). Ergosterol is an important constituent of the cell membranes of higher fungi, and as such, it correlates fairly well with fungal biomass. Using qPCR assays, the total bacterial and fungal communities were assessed, and no significant differences in fungal and bacterial DNA concentrations were observed, neither between invaded and non-invaded habitats nor between the two habitat types (Table 3). For further verification, two single-copy fungal markers (TUB2, a beta-tubulin) and TEF1, translation elongation factor 1-alpha) were tested, and no significant differences between invaded and uninvaded plots were observed (data not shown). 

Keeping in mind that ergosterol measurements predominantly reflect the presence of Ascomycota and Basidiomycota, representatives of two major distal clades within the kingdom Fungi [21], these phyla were quantified separately. In the riparian vegetation habitats, a trend was observed of Ascomycota having a higher DNA concentration in *S. gigantea*-invaded plots (F_1,50_ = 3.31, *p* = 0.0748; Table 3, Figure 1). A similar invasion effect was observed when both habitats were analysed together (F_1,50_ = 3.34, *p* = 0.0738; Table 2). The mean DNA concentration of Basidiomycota on sandy soils was about three times higher than the DNA concentration in the river clay soils (F_1,6_ = 10.83, *p* = 0.0166; Table 2). The DNA concentrations of Basidiomycota did not differ between invaded and uninvaded plots (Table 2 and Table 3, Figure 1). In addition, Chytridiomycota were measured, being a fungal phylum that uses cholesterol instead of ergosterol as its major sterol, but no differences were observed between giant goldenrod-invaded and uninvaded plots. Comparison of Chytridiomycota between the two major habitats revealed no difference in DNA concentrations. 

The overall bacterial DNA concentration tended to be slightly higher in *S. gigantea*-invaded plots (F_1,50_ = 3.29, *p* = 0.0759; Table 2, Figure 1), but there were no significant effects of habitat type (Table 3, Figure 1). 

### 3.3. Two Fungivorous Nematode Families Benefited from S. gigantea-Induced Increase in Fungal Biomass

The total nematode abundance and density of the three fungivorous nematode taxa that were commonly present in both the Pleistocene sand and river clay locations were analysed to study the belowground impact of *S. gigantea* on the next trophic level of the soil food web. Representatives of the families Aphelenchidae, Aphelenchoididae, and the genus *Diphtherophora* were used to determine whether and if so how the observed increase in biomass of higher fungi and unchanged fungal DNA levels were reflected in the local fungivorous nematode community. 

Dominance by giant goldenrod did not affect the total nematode density (Table 2, Figure 1). Total nematode density (per 100 g dry soil) only differed significantly between habitats. The riparian sites had an estimated mean nematode abundance about two times higher than in semi-natural grassland soils (F_1,6_ = 26.69, *p* = 0.0021; Table 2). Both Aphelenchidae (F_1,50_ = 9.96, *p* = 0.0027) and Aphelenchoididae (F_1,50_ = 8.44, *p* = 0.0054; Table 2, Figure 1) were more abundant in *S. gigantea*-invaded plots than in uninvaded plots. A significant interactive effect between habitat type and invasion status (F_1,50_ = 6.92, *p* = 0.0113) was observed for Aphelenchoididae indicating that the response to invasion was dependent on habitat type, whereas this interactive effect between habitat type and invasion status was not significant for Aphelenchidae (F_1,50_ = 3.16, *p* = 0.0814; Table 2). 

For both fungivorous nematode taxa, the effect of *S. gigantea* was only seen in the riparian habitats. As compared to the uninvaded plots, Aphelenchidae densities were three times higher in invaded plots in riparian habitats (F_1,50_ = 11.30, *p* = 0.0015; Table 3, Figure 1). Similarly, the estimated densities of Aphelenchoididae were around four times higher in *S. gigantea*-invaded riparian plots as compared to the uninvaded neighbouring plots (F_1,50_ = 14.23, *p* = 0.0004; Table 3, Figure 1). Giant goldenrod stands did not affect the abundance of representatives of the genus *Diphtherophora* (Table 2 and Table 3, Figure 1). 

### 3.4. Fungal Indicator Taxa Associated with Invasive S. gigantea

Invasion by *S. gigantea* resulted in a local increase in fungal biomass, but not in total fungal DNA. This remarkable observation was investigated in more detail by comparing the composition of the communities. As a crude measure for fungal DNA content, we compared the number of primary reads per sample. Whereas soil samples from uninvaded plots gave rise to ≈95,000 (SD 39,000) reads per sample, on average ≈102,000 reads (SD 37,000) were generated from samples from invaded plots. There was no significant difference in number of reads per sample between invaded and uninvaded plots. Although it is hard to compare qPCR data with Illumina reads, the sequencing data confirm the absence of a difference in fungal DNA contents between *S. gigantea*-invaded and uninvaded plots. 

PERMANOVA on Bray-Curtis dissimilarity profiles identified “habitat type” (riparian vegetation vs. natural grassland) as the main factor responsible for the difference in fungal composition (Table 4). This factor explained ≈27% of the overall variance. The second most informative variable was “study site” with an R^2^ value of 0.23. This is the variation in fungal communities between the various sampling sites within a habitat type. Against the substantial background variation caused by habitat type and study site, a clear invasion effect still could be discerned. This effect explained a low but significant percentage of the overall variation (1.7%). As can be seen in Table 4, analyses of fungal communities for the two habitat types separately revealed significant effects. The effect of plant invasion on fungal assemblages was more pronounced in the semi-natural grasslands (*p* = 0.001 for semi-natural grassland, and *p* = 0.01 for riparian vegetation). 

Linear discriminant analysis Effect Size (LEfSe) analysis allowed us to determine which fungal taxa contributed most to observed differences between *S. gigantea* invaded and uninvaded plots. With an LDA threshold of >2, the families Cladosporiaceae, Teratosphaeriaceae (both Ascomycota), Glomeraceae (Glomeromycota), and Kondoaceae (Basidiomycota) were shown to be more abundant in plots invaded with *S. gigantea* (Figure 2A). Analysis per habitat type revealed that only the Cladosporaceae were present in higher densities in both habitat types in invaded plots (Figure 2B,C). A higher abundance of members of the family Cucurbitariaceae (Ascomycota) was shown to be characteristic for the uninvaded plots.

## 4. Discussion

Our results showed quantitative and qualitative shifts in the fungal community brought about by the invasive plant species *S. gigantea*. Consistently, i.e., during multiple growth stages, over multiple years, and at multiple locations, invasive giant goldenrod was shown to induce a local increase in fungal biomass [19,20]. Qualitative characterization of the fungal assemblages revealed that *S. gigantea* invasion was accompanied by a local increase in abundance of members of the families Cladosporiaceae and Glomeraceae, and a decreased presence of Cucurbitariaceae followed by an increase of the fungivorous nematode lineages Aphelenchoididae and Aphelenchidae. The interpretations of our results are presented below.

### 4.1. Apparent Discrepancy between Results from Independent Fungal Biomass Markers 

Notably, the observed increase in ergosterol in *S. gigantea*-invaded plots was not accompanied by a comparable local augmentation of the total fungal DNA (Figure 1). In the case of the phylum Ascomycota, a trend towards more rDNA in invaded plots was detected (Table 3, *p* = 0.0748). With regard to the Basidiomycota, the apparent low density (Table 3) might be an underestimation as relatively high rDNA copy representatives were used to generate the calibration lines [27,42].

Ergosterol is a frequently used marker for the assessment of fungal biomass in soil. This sterol is found in all Ascomycota and most Basidiomycota. Several representatives of the Zygomycota harbour ergosterol in their membranes as well, but this sterol is absent in the more basal fungal lineages [21]. Using cultures of six non-basal fungal species [43] showed a tight correlation between ergosterol content and fungal biomass. Given that the local fungal communities were dominated by later diverging divisions such as Ascomycota and Basidiomycota [44], ergosterol is considered to be an adequate marker for fungal biomass at these locations. 

With regard to the use of rDNA as a marker for fungal biomass, it should be mentioned that genome-based surveys revealed considerable variation in rDNA copy numbers. Nevertheless, some phylum-specific characteristics have been observed. The average number of rDNA copies for Ascomycota is around 50 and shows limited variation. Basidiomycota harbor about twice as many rDNA copies, and this is accompanied by substantial copy number variation among its members [42]. Hence, rDNA copy numbers can only be used to assess fungal biomass in cases where there are no major differences between the community composition of the samples. The rDNA-based estimation of the Ascomycota is likely to be more accurate than the estimation of the Basidiomycota biomass. Although both ergosterol and rDNA copy number have their limitations as fungal biomass markers in soil, the comparisons of data from adjacent plots from the same habitat are probably valid. Fungi are known to be more plasticity with regard to the biomass:DNA ratio than many other organismal groups [45]. This is the result of the cellular organization of fungi. The hyphal compartmentalization of fungi might be impaired by the partial or complete removal of septa, cross walls separating the fungal cells [46]. Thus, growth of the mycelial network does not necessarily have to be accompanied by a comparable increase in the number of nuclei. Therefore, the difference in outcome between the two types of markers (biochemical or DNA-based) might be attributable to an increase in the fungal biomass:DNA ratio. Further research is required to investigate this hypothesis. 

### 4.2. The Habitat (In)Dependent Impact of S. gigantea on Fungivorous Nematode Lineages 

Due to the apparently contradictory results obtained by the two types of fungal biomass markers, the effects of *S. gigantea* on major representatives of the next trophic level, fungivorous nematodes, were checked. Two out of three lineages of fungivores, Aphelenchoididae and Aphelenchidae, both present in the riparian zone and in the semi-natural grassland sites, were shown to be stimulated in the presence of giant goldenrod, whereas the third lineage, the genus *Diphtherophorida*, was unaffected by this invasive plant species. Moreover, the effects in the riparian habitats were much more pronounced than the effects in the sandy locations. 

While the increase in Aphelenchoididae as a result of giant goldenrod invasion was found previously [19], the increase in Aphelenchidae was new. As nematode densities in these fields fluctuate during the season [28], it is conceivable that the *S. gigantea*-induced increase in Aphelenchidae is only noticeable late in the season. A recent, elaborate study found many Aphelenchoididae species as well as *Aphelenchus avenae* (Aphelenchidae) to be positively correlated with invasive *S. gigantea* plots [47]. This strengthens our findings that both lineages are stimulated by *S. gigantea* invasion. 

A boost of Aphelenchoididae and Aphelenchidae was observed in river clay soil, and a non-significant increase was seen in sandy soils (e.g., Figure 1). This difference in response might be explained by the soil texture-dependent species representation for each of these two families. Apparently, the *Aphelenchoides* species present in the river clay soil could benefit more from the local increase in fungal biomass than the *Aphelenchoides* species present in the sandy soils. We propose the same line of reasoning for the Aphelenchidae genera *Aphelenchus* and *Paraphelenchus*. Representatives of the genus *Diphtherophora* were unaffected by the presence of giant goldenrod, which could be caused by a difference in food preference between the lineages [24,48,49]. 

The fungivorous nematode densities reported in this study are relatively low. This could be a late season sampling effect. This effect has little impact on the current analyses as differences between uninvaded and invaded plots are considered rather than absolute changes. 

### 4.3. Effect of Habitat-Characteristic Abiotic Differences between Habitat-Type

Despite the differences in floristic composition, soil type and land use history between the riparian zone and the semi-natural grasslands, the overall biotic impact of giant goldenrod induced similar overall invasion effects. Nevertheless, Basidiomycota were more abundant in semi-natural grasslands than in the riparian zones (*p* = 0.017, Table 2). This could relate to a substantial pH difference. Whereas riparian zones had a relatively neutral pH of 7.5, semi-natural grasslands had a pH nearly 2 units lower. As compared to bacteria, pH windows for optimal growth are wider for fungi [50]. In a more recent study, fungal communities in arctic soils with a pH range of over 2.5 units were investigated [51]. In the most acidic sites, Basidiomycota showed a higher relative abundance as compared to sites with more basic soils. The higher abundance in Basidiomycota in semi-natural grasslands did not result in a significant change in either of the fungivorous nematode lineages. 

### 4.4. Fungal Indicator Taxa Related to Invasion with S. gigantea

As shown in Figure 2, the Ascomycete family Cladosporiaceae was one of the main families responsible for the *S. gigantea*-induced shift in fungal community composition. A closer look at the Cladosporiaceae OTUs revealed that *Cladosporium* was the dominant genus within this family. 

*Cladosporium* is a fairly speciose genus. It comprises 189 species that are mostly saprotrophic; in addition, it also harbours some plant pathogens [52]. Although in its native range, leaves of showy goldenrod (*Solidago speciosa*) were shown to be infested by *Cladosporium asterum,* causing brown rust pustules (website Missouri Botanical Garden, USA), no information was found on *Cladosporium* being an important pathogen of *S. gigantea* in Europe. On the other hand, plant growth-promoting characteristics of members of the genus *Cladosporium* could be marked as a possible benefit for *S. gigantea* associated with fostering *Cladosporium* in its rhizosphere. Both *Cladosporium sphaerospermum* and *Cladosporium* sp. MH-6 were found to produce and release several types of gibberellins with plant growth-promoting characteristics [53,54]. 

Glomeraceae was the second most indicative family regarding the impact of *S. gigantea* on the local fungal community. Glomeraceae is a family of arbuscular mycorrhizal fungi and members of this family colonize the roots of a wide range of vascular plants including *S. gigantea* [55]. Vallino et al. [56] characterized the AMF colonization of *S. gigantea* outside its native range, and identified *Glomus*, a genus belonging to the Glomeraceae, as the dominant root colonizing AMF [56]. This result underlines that invasive *S. gigantea* can recruit local AMF and establish such a successful interaction that it ends up as one of the main fungal taxa typifying the community shift that was brought about by the invasive plant species. 

The Glomeraceae were a specific indicator for invaded plots in semi-natural grassland. Subsequently, we also found a significant increase of SOM in these invaded plots. As AMF-derived carbon in itself may be a significant component of SOM [57,58], it also has been suggested that AMF symbiosis increases soil C by reducing the decomposition of SOM [59]. Moreover, AMF could also alter SOM by affecting other saprobic soil microbiota [13]. Unfortunately, no sequencing data from the bacterial community were available to investigate this in more detail. 

In both habitat types, uninvaded plots were characterized by an increased presence of Cucurbitariaceae, just like the Cladosporiaceae family that belongs to the class Dothideomycetes. Little is known about the ecology of Cucurbitariaceae. Its members are known as saprobes on relatively recalcitrant organic materials such as wood, bark, and leaves [60].

High-throughput sequencing revealed multiple fungal families as indicative for invasion. In the previous section we hypothesized that this difference in response could have been caused by a difference in food preference between the lineages. Interestingly, *Cladosporium* was identified as a moderate feeding source for *Aphelenchoidides* sp [49], which is in line with our observations. Unfortunately, little is known about food preferences of fungivorous nematode taxa. The indicator families observed in this research could be an interesting starting point for research targeting nematode feeding preferences. The fungal families boosted by *S. gigantea* in a riparian habitat could be especially informative as Aphelenchidae and Aphelenchoididae showed a remarkable increase in these soils.

## 5. Conclusions

This study shows that *S. gigantea* invasion has a structural impact on the belowground soil community by increasing the fungal biomass independent of the sampling moment, sampling year, or habitat. The increase of fungal biomass is reflected in the next trophic level by a boost of two independent lineages of fungivorous nematodes, Aphelenchidae and Aphelenchoididae.

The ergosterol-based observation of the increase in the fungal biomass a *S. gigantea*, could not be confirmed by DNA markers. Both qPCR-based assessment of total fungal DNA as well as the characterization of the fungal communities on the basis of variable 18S rDNA regions did not reveal a difference in fungal DNA content between *S. gigantea*-invaded and uninvaded plots. The apparent discrepancy might be attributable to a change in the DNA:biomass ratio. High throughput sequencing of the 18S rDNA regions V7-V8 revealed a stimulation of Cladosporiaceae and Glomeraceae and a suppression of the Cucurbitariaceae as a result of giant goldenrod invasion. Further investigation into the nature of these community shifts might elucidate the observed change in fungal DNA:biomass ratio induced by the invasive plant species *S. gigantea*.

## Figures and Tables

**Figure 1 microorganisms-09-00437-f001:**
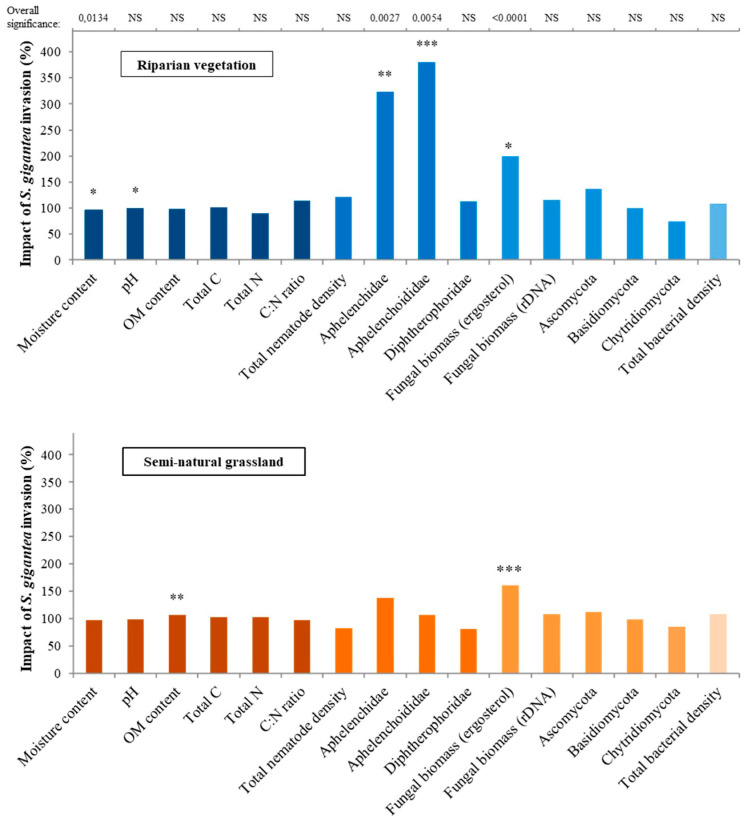
Impact of *S. gigantea* invasion in riparian vegetation (top) and semi-natural grassland habitat. The impact of *S. gigantea* invasion on the *y*-axis was calculated by dividing estimated means (Table 3) from invaded plots by estimated means from un-invaded plots and expressed as a percentage. Impacts are shown for the 6 abiotic variables, the total nematode density, densities of three fungivorous nematodes, total fungal density, densities of three fungal phyla and the total bacterial density (no invasion impact = 100%). Asterisks indicate significant differences (* *p* < 0.05, ** *p* < 0.01, *** *p* < 0.001) between invaded and un-invaded plots per habitat type. Variables showing an overall significant invasion effect, for both habitats together, are indicated by a grey-shaded background. Corresponding *p*-values are shown at the top part of the figure (NS = not significant). Riparian vegetation habitats included 3 study sites and 24 plot-pairs, while semi-natural grasslands included 5 study sites and 28 plot-pairs.

**Figure 2 microorganisms-09-00437-f002:**
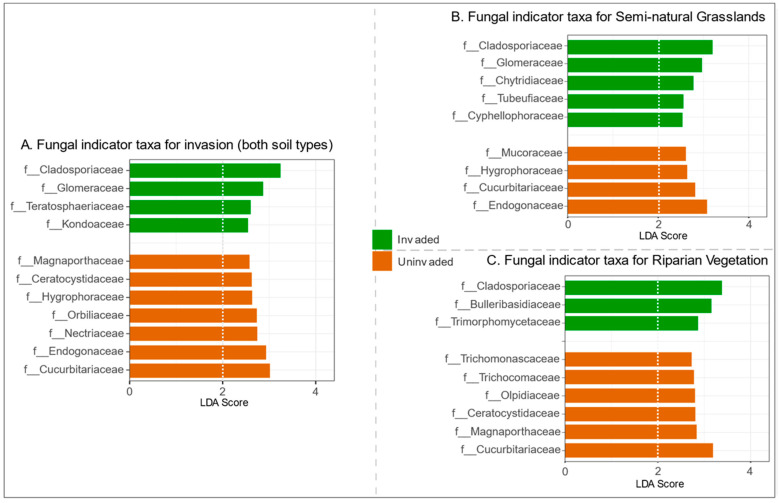
Discriminant fungal families indicated by LEfSe analysis (LDA threshold of 2) resulting from in-vaded (green) and un-invaded (brown) soils by Solidago gigantea.

**Table 1 microorganisms-09-00437-t001:** Eight study sites located in riparian zones and in semi-natural grassland habitats are indicated below. Although “Blauwe Kamer” is one riparian study site, samples were collected from two distinct areas within the nature reserve (1 and 2). Riparian zones are characterized by river clay soils, whereas the semi-natural grassland sites were located on Pleistocene sandy soils.

Habitat Type	Study Site	Soil Type	Coordinates	Year of *S. gigantea* Introduction	Number of Plots Pairs
**Riparian zone**	Millingerwaard	River clay	51°51′58.11″ N 6°00′35.47″ E	~1950	8
	Ewijkse plaat	River clay	51°52′47.36″ N 5°44′52.17″ E	~1950	8
	Blauwe Kamer				
	*West*	River clay	51°56′40.22″ N 5°36′19.90″ E	after 1950	4
	*East*	River clay and sand	51°56′32.56″ N 5°37′09.54″ E	after 1950	4
**Semi-natural grassland**	Dennenkamp	Pleistocene sand	52°01′45.64″ N 5°47′53.50″ E	1982	8
	Plantage Willem III	Pleistocene sand	51°58′48.62″ N 5°31′08.47″ E	1995	8
	Hollandseweg	Pleistocene sand	51°58′49.89″ N 5°40′59.84″ E	before 2005	4
	Scheidingslaan	Pleistocene sand	51°58′28.60″ N 5°41′55.40″ E	unknown	4
	Reijerscamp	Pleistocene sand	52°00′47.49″ N 5°46′08.64″ E	2006	4

**Table 2 microorganisms-09-00437-t002:** Main effects of habitat type, invasion, and their interaction (*) for the different abiotic and biotic variables analysed. F-test Fdf values and corresponding p-values obtained from the mixed models are shown for each variable. Total C and N contents are expressed in g/kg dry soil. Total nematode density, Aphelenchidae, Aphelenchoididae, and Diphtherophora are expressed in numbers (#) per 100 g dry soil. Total fungal density, Ascomycota, Basidiomycota, Chytridiomyco-ta, and total bacterial density are expressed in μg DNA per 100 g dry soil. Fungal biomass is ex-pressed as mg ergosterol kg-1 soil. The degrees of freedom (Crowther et al.) for Diphtherophora are lower than for the other variables, since this taxon was not present at two study sites (Scheidings-laan and Reijerscamp). Regarding invasion and interaction effects, the df for total C, N, and C:N ratio are lower since samples were pooled together per study site. Significant p-values (< 0.05) are indicated in bold.

	Habitat Type			Invasion			Habitat Type * Invasion		
Abiotic Variables	F_df_		*p*-Value	F_df_		*p*-Value	F_df_		*p*-Value
**Moisture content (%)**	F_1,6_	2.02	0.2052	F_1,50_	6.58	**0.0134**	F_1,50_	0.93	0.3391
**pH**	F_1,6_	92.22	**<0.0001**	F_1,50_	2.48	0.1213	F_1,50_	0.19	0.6661
**Organic matter (OM) content (%)**	F_1,6_	0.13	0.7343	F_1,50_	2.87	0.0965	F_1,50_	4.74	**0.0341**
**Total C (g/kg)**	F_1,6_	1.38	0.2848	F_1,6_	0.05	0.8281	F_1,6_	0.00	0.9562
**Total N (g/kg)**	F_1,6_	0.57	0.4803	F_1,6_	0.15	0.7090	F_1,6_	0.60	0.4692
**C:N ratio**	F_1,6_	0.73	0.4267	F_1,6_	0.43	0.5378	F_1,6_	1.00	0.3568
Biotic variables									
**Total nematode density (#)**	F_1,6_	26.69	**0.0021**	F_1,50_	0.00	0.9758	F_1,50_	3.24	0.0780
**Aphelenchidae (#)**	F_1,6_	0.02	0.8962	F_1,50_	9.96	**0.0027**	F_1,50_	3.16	0.0814
**Aphelenchoididae (#)**	F_1,6_	0.65	0.4507	F_1,50_	8.44	**0.0054**	F_1,50_	6.92	**0.0113**
***Diphtherophora (#)***	F_1,4_	0.08	0.7930	F_1,42_	0.02	0.9007	F_1,42_	0.29	0.5945
**Fungal biomass (mg erg/kg)**	F_1,6_	3.63	0.1055	F_1,48_	21.97	**<0.0001**	F_1,48_	0.72	0.3990
**Fungal DNA (μg)**	F_1,6_	0.75	0.4184	F_1,50_	2.50	0.1203	F_1,50_	0.17	0.6846
**Ascomycota (μg)**	F_1,6_	0.06	0.8187	F_1,50_	3.34	0.0738	F_1,50_	0.71	0.4027
**Basidiomycota (μg)**	F_1,6_	10.83	**0.0166**	F_1,50_	0.00	0.9637	F_1,50_	0.01	0.9370
**Chytridiomycota (μg)**	F_1,6_	1.07	0.3405	F_1,50_	1.25	0.2693	F_1,50_	0.12	0.7340
**Total bacterial density (μg)**	F_1,6_	0.05	0.8360	F_1,50_	3.29	0.0759	F_1,50_	0.02	0.8826

**Table 3 microorganisms-09-00437-t003:** Estimated mean response and associated 95% confidence intervals of the soil characteristics analyzed for plots invaded and uninvaded by *S. gigantea* in two habitat types. Values were obtained from “least squares means” outputs of mixed models fitted to the variables. For both habitat types, riparian vegetation, and semi-natural grassland, ect. mean responses are shown for plots invaded and uninvaded by *S. gigantea*. Riparian vegetation habitats contained 24 plot-pairs in total, while semi-natural grasslands contained 28 plot-pairs in total. Values for OM content, total C (g/kg dry soil), total N (g/kg dry soil), Aphelenchidae, Aphelenchoididae, *Diphtherophora*, total fungi, Ascomycota, Basidiomycota, Chytridiomycota, and total bacteria were back transformed from logarithmic values to the original scale. Aphelenchidae, Aphelenchoididae, and *Diphtherophora* are expressed in numbers (#) per 100 g dry soil. For *Diphtherophora*, 8 plot-pairs from semi-natural grasslands were excluded from analysis. Fungal biomass is expressed as mg ergosterol per kg soil. Total fungal density, Ascomycota, Basidiomycota, Chytridiomycota, and total bacterial density are expressed in μg DNA per 100 g dry soil. Significant *p*-values (< 0.05) are indicated in bold.

	Riparian Vegetation (*n* = 24 Plot-Pairs)							Semi-Natural Grassland (*n* = 28 Plot-Pairs)						
	Invaded (*n* = 24)			Uninvaded (*n* = 24)				Invaded (*n* = 28)			Uninvaded (*n* = 28)			
	Lower	Est. mean	Upper	Lower	Est. mean	Upper	*p*-Value	Lower	Est. mean	Upper	Lower	Est. mean	Upper	*p*-Value
Abiotic variables														
**Moisture content (%)**	15.9	**20.7**	25.6	16.7	**21.5**	26.4	**0.0199**	12.8	**16.6**	20.4	13.2	**17.0**	20.7	0.2445
**pH**	7.33	**7.46**	7.59	7.37	**7.50**	7.63	**0.0197**	5.21	**5.58**	5.96	5.28	**5.65**	6.02	0.3059
**OM content (%)**	4.2	**5.9**	8.3	4.2	**5.9**	8.4	0.7429	4.3	**5.7**	7.4	4.0	**5.3**	6.9	**0.0063**
**Total C (g/kg)**	17.0	**31.2**	57.0	16.8	**30.6**	56.0	0.9181	13.9	**22.1**	35.3	13.5	**21.5**	34.4	0.8243
**Total N (g/kg)**	0.9	**1.7**	3.1	1.0	**1.9**	3.4	0.4895	0.9	**1.5**	2.3	0.9	**1.4**	2.2	0.7663
**C:N ratio**	12.9	**19.1**	25.2	10.6	**16.8**	22.9	0.3366	10.5	**15.2**	20.0	10.9	**15.7**	20.5	0.7879
Biotic variables														
**Aphelenchidae (#)**	0.4	**1.6**	6.0	0.1	**0.5**	2.0	**0.0015**	0.4	**1.2**	3.6	0.3	**0.9**	2.7	0.3156
**Aphelenchoididae (#)**	1.4	**4.3**	13.0	0.3	**1.1**	3.6	**0.0004**	1.6	**4.0**	9.8	1.5	**3.7**	9.2	0.8404
***Diphtherophora (#)***	0.4	**0.9**	1.9	0.4	**0.8**	1.7	0.7621	0.4	**0.9**	2.0	0.5	**1.1**	2.4	0.6564
**Fungal biomass (mg erg/kg)**	1.07	**1.75**	2.85	0.54	**0.88**	1.43	**0.0199**	1.74	**2.66**	3.97	1.22	**1.65**	2.76	**<0.0001**
**Fungal DNA (μg)**	220	**299**	405	191	**259**	351	0.1814	264	**340**	438	243	**313**	403	0.3925
**Ascomycota (μg)**	26.1	**40.2**	61.9	19.1	**29.3**	45.2	0.0748	26.9	**38.7**	55.6	24.0	**34.5**	49.5	0.4730
**Basidiomycota (μg)**	1.6	**2.8**	5.1	1.6	**2.8**	5.0	0.9818	5.4	**8.8**	14.3	5.5	**9.0**	14.5	0.9270
**Chytridiomycota (μg)**	0.4	**0.9**	1.8	0.6	**1.2**	2.4	0.3250	0.3	**0.6**	1.2	0.4	**0.7**	1.4	0.5709
**Total bacterial density (μg)**	3598	**4929**	6754	3303	**4526**	6201	0.1875	3654	**4696**	6034	3399	**4367**	5612	0.2263

**Table 4 microorganisms-09-00437-t004:** Summary PERMANOVAs on Bray-Curtis dissimilarity profiles of the fungal biome for the main effect and each habitat type. The effects of the following variables on the quantitative taxonomic composition of fungi were tested: Habitat type (riparian vegetation, and semi-natural grassland), Study site (*n*_rv_ = 3, *n*_sng_ = 5), Invasion (invaded/uninvaded), and the interactions (*) between Invasion and Habitat Type or Study Site. Differences were considered significant if *p* < 0.01. *p* = probability associated with the Pseudo F statistic. Significant *p*-values are indicated in bold.

	Main Effect		Riparian Vegetation		Semi-Natural Grassland	
	*R* ^2^	*p*	*R* ^2^	*p*	*R* ^2^	*p*
**Habitat Type**	0.274	**0.0001**				
**Study Site**	0.227	**0.0001**	0.257	**0.001**	0.356	**0.001**
**Invasion**	0.017	**0.005**	0.037	**0.010**	0.033	**0.001**
**Study Site * Invasion**	0.030	0.463	0.030	0.532	0.051	0.327
**Habitat * Invasion**	0.009	0.082				

## Data Availability

The raw sequences were submitted to the NCBI Sequence Read Archive (SRA) database under study accession numbers PRJNA563313.

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
