# Peer review of "Characterization of the Habitat- and Season-Independent Increase in Fungal Biomass Induced by the Invasive Giant Goldenrod and Its Impact on the Fungivorous Nematode Community"

_microorganisms, 2021, doi:10.3390/microorganisms9020437_

Round 1

Reviewer 1 Report

  1. Comment: Determining organic matter content by weight loss has been demonstrated to overestimate C for soil samples with clay minerals containing significant amounts of interlayer water. The authors might consider this in relation to their C and OM results.
  2. In the Materials & Methods section, the reader is referred to "table 1" for information on primers, PCR conditions etc., but this table was not included in the file that I received.
  3. The authors should carefully check citations in the text, see for example ref 34 on page 5 (Materials & Methods section).
  4. The authors should check table numbers in the text.

Author Response

Response to reviewer 1

Dear reviewer 1, thank you for your time reviewing the manuscript. All changes are highlighted in the main manuscript by track changes. Due to the amount of changes, we would advise to read the manuscript with ‘Simple Markup’ in the review - tracking plane and only switch to ‘All Markup’ in order to see specific changes or search corresponding line numbers.

  1. Determining organic matter content by weight loss has been demonstrated to overestimate C for soil samples with clay minerals containing significant amounts of interlayer water. The authors might consider this in relation to their C and OM results.

We thank reviewer 1 for making us aware of the possible overestimation of C for soil samples with clay minerals. We added this to our results. Since we are mainly focused on comparing invaded with un-invaded plots, we believe this is not changing our conclusions.

Line 299: “It should be noted that determining OM content by weight loss as used here might have overestimated the OM% for the riparian samples with a high clay content [41]

[41] Barille-Boyer, A. L., Barille, L., Massé, H., Razet, D., & Heral, M. (2003). Correction for particulate organic matter as estimated by loss on ignition in estuarine ecosystems. Estuarine, Coastal and Shelf Science, 58(1), 147-153.

  1. In the Materials & Methods section, the reader is referred to "table 1" for information on primers, PCR conditions etc., but this table was not included in the file that I received.

We thank the reviewer for pointing this out. Due to adjusting the references to a numerical system we accidentally referred to the wrong table. This should have been Table 3 of Harkes et al [29]. This is now corrected in the MS, see line 175.

  1. The authors should carefully check citations in the text, see for example ref 34 on page 5 (Materials & Methods section).

Done, all citations have been checked and double citations have been removed throughout the manuscript. Citation in text should now be linked unequivocally to the reference list.

  1. The authors should check table numbers in the text.

This has been checked. All tables are now properly referred to in the text.

Reviewer 2 Report

Paper presents interesting results on the impact of invasive species on soil nematodes. There are several inconsistencies and redundancies that make the paper very hard to follow. The paper needs careful revision to address the following:
1. Genus name and epithet should be italicized. There are inconsistencies throughout the text.
2. Need to follow the format for the intext citations.
3. Need to be consistent, Fig and Figure are written randomly.
4. Need to follow the format for the tables.
5. Methods, Discussion, and Conclusion sections should be rewritten entirely be more cohesive and avoid redundancies.
6. The reference section has two numberings and they are inconsistent.

Author Response

Response to reviewer 2

Paper presents interesting results on the impact of invasive species on soil nematodes. There are several inconsistencies and redundancies that make the paper very hard to follow. The paper needs careful revision to address the following

Dear reviewer 2, thank you for your time reviewing the manuscript. We have contacted a professional proof reader (a native speaker) to improve the linguistic quality of the MS text as suggested. All changes are highlighted in the main manuscript by track changes. Due to the number of changes, we suggest to read the manuscript with ‘Simple Markup’ in the ‘review – tracking’ plane and only switch to ‘All Markup’ in order to see specific changes or search corresponding line numbers.

  1. Genus name and epithet should be italicized. There are inconsistencies throughout the text.

We checked all genus and species names and corrected them where needed. For example lines: 17, 29, 44, 228, 358, 451, 505 and genus names in all references.

  1. Need to follow the format for the intext citations.

Done, all citations have been checked and corrected. For example lines: 336, 509, 574, 592, 596.

  1. Need to be consistent, Fig and Figure are written randomly.

This is corrected and is written as ‘Figure’ throughout the MS.

  1. Need to follow the format for the tables.

We have changed all tables to the same format, font size and colour scheme in order to improve consistency of the manuscript.

  1. Methods, Discussion, and Conclusion sections should be rewritten entirely be more cohesive and avoid redundancies.

We have reviewed and adjusted all of the mentioned sections  in the revised manuscript.  The materials and methods have been textually and content-wise streamlined to improve both the flow and the readability. In the discussion and conclusion, redundant information has been deleted, and phrasing has been adjusted on numerous places to make our lines of reasoning more logical, and easier to read.

  1. The reference section has two numberings and they are inconsistent.

This disordered reference section has been thoroughly checked and should not contain any double numbers nor missing references anymore. 

Round 2

Reviewer 2 Report

The manuscript looks much better with the revision. There are still some grammatical errors in the text. Authors carefully revise the manuscript and make corrections.